# DNA Methylation in the Diagnosis of Monogenic Diseases

**DOI:** 10.3390/genes11040355

**Published:** 2020-03-26

**Authors:** Flavia Cerrato, Angela Sparago, Francesca Ariani, Fulvia Brugnoletti, Luciano Calzari, Fabio Coppedè, Alessandro De Luca, Cristina Gervasini, Emiliano Giardina, Fiorella Gurrieri, Cristiana Lo Nigro, Giuseppe Merla, Monica Miozzo, Silvia Russo, Eugenio Sangiorgi, Silvia M Sirchia, Gabriella Maria Squeo, Silvia Tabano, Elisabetta Tabolacci, Isabella Torrente, Maurizio Genuardi, Giovanni Neri, Andrea Riccio

**Affiliations:** 1Dipartimento di Scienze e Tecnologie Ambientali Biologiche e Farmaceutiche, Università degli Studi della Campania “Luigi Vanvitelli”, 81100 Caserta, Italy; flavia.cerrato@unicampania.it (F.C.); angela.sparago@unicampania.it (A.S.); 2Genetica Medica, Università di Siena, 53100 Siena, Italy; ariani2@unisi.it; 3UOC Genetica Medica, Fondazione Policlinico Universitario A. Gemelli IRCCS, 00168 Rome, Italy; fulvia.bru@gmail.com (F.B.); fiorella.Gurrieri@unicatt.it (F.G.); eugenio.sangiorgi@unicatt.it (E.S.); maurizio.genuardi@unicatt.it (M.G.); 4Fondazione Policlinico Universitario A. Gemelli IRCCS, Istituto di Medicina Genomica, Università Cattolica del Sacro Cuore, 00168 Rome, Italy; elisabetta.tabolacci@unicatt.it (E.T.); giovanni.neri43@gmail.com (G.N.); 5Laboratorio di Citogenetica Medica e Genetica Molecolare, Centro di Ricerche e Tecnologie Biomediche IRCCS, Istituto Auxologico italiano, 20149 Milan, Italy; luciano.calza@gmail.com (L.C.); s.russo@auxologico.it (S.R.); 6Dipartimento di Ricerca Traslazionale e delle Nuove tecnologie in Medicina e Chirurgia, Laboratorio di Genetica Medica, Università di Pisa, 56126 Pisa, Italy; fabio.coppede@med.unipi.it; 7Divisione di Genetica Medica, Fondazione IRCCS Casa Sollievo della Sofferenza, San Giovanni Rotondo, 71013 Foggia, Italy; a.deluca@css-mendel.it (A.D.L.); g.merla@operapadrepio.it (G.M.); g.squeo@operapadrepio.it (G.M.S.); i.torrente@css-mendel.it (I.T.); 8Genetica Medica, Dipartimento di Scienze della Salute, Università degli studi di Milano, 20146 Milan, Italy; cristina.gervasini@unimi.it (C.G.); silvia.sirchia@unimi.it (S.M.S.); 9Laboratorio di Medicina Genomica UILDM, Fondazione Santa Lucia, 00179 Rome, Italy; emiliano.giardina@uniroma2.it; 10Dipartimento di Biomedicina e Prevenzione, Università degli Studi di Roma “Tor Vergata”, 00133 Rome, Italy; 11Genetica Medica, Università Campus Bio-Medico, 00128 Rome, Italy; 12Laboratori Centrali, EO Ospedali Galliera, 16128 Genova, Italy; cristiana.lo.nigro@galliera.it; 13Dipartimento di Fisiopatologia Medico-chirurgica e dei Trapianti, Genetica Medica, Università degli Studi di Milano, 20122 Milan, Italy; monica.miozzo@unimi.it (M.M.); silvia.tabano@unimi.it (S.T.); 14Fondazione IRCCS Ca’ Granda Ospedale Maggiore Policlinico, Milano, Università degli Studi di Milano, 20122 Milan, Italy; 15Self Research Institute, Greenwood Genetic Center, Greenwood, SC 29646, USA; 16Istituto di Genetica e Biofisica “Adriano Buzzati Traverso” Consiglio Nazionale delle Ricerche, 80131 Naples, Italy

**Keywords:** DNA methylation, genetic testing, high-throughput analysis, epi-signatures, developmental delay/intellectual disability disorders, imprinting disorders, hereditary tumors, neuromuscular diseases, prenatal diagnosis

## Abstract

DNA methylation in the human genome is largely programmed and shaped by transcription factor binding and interaction between DNA methyltransferases and histone marks during gamete and embryo development. Normal methylation profiles can be modified at single or multiple loci, more frequently as consequences of genetic variants acting in cis or in trans, or in some cases stochastically or through interaction with environmental factors. For many developmental disorders, specific methylation patterns or signatures can be detected in blood DNA. The recent use of high-throughput assays investigating the whole genome has largely increased the number of diseases for which DNA methylation analysis provides information for their diagnosis. Here, we review the methylation abnormalities that have been associated with mono/oligogenic diseases, their relationship with genotype and phenotype and relevance for diagnosis, as well as the limitations in their use and interpretation of results.

## 1. Introduction 

DNA methylation is by far the most abundant modification of the human genome. It is mostly present at cytosines of CpG dinucleotides, and the resulting CpG symmetry on the complementary helices of DNA helps the process of DNA methylation maintenance through cell division by the methyltransferase DNMT1 [1,2,3]. In mammals, DNA methylation is largely reprogrammed during gametogenesis and early embryo development [4,5]. Two waves of extensive erasure involving both passive and active mechanisms occur in the primordial germ cells and pre-implantation embryo. De novo DNA methylation catalyzed by the methyltransferases DNMT3A, DNMT3B and DNMT3L occurs with differential kinetics and patterns during male and female gametogenesis and within cell lineage specification in post-implantation development [4,5]. The distribution of CpG methylation (mCpG) along the human genome is not uniform. While most of the genome is CpG-poor and methylated in differentiated somatic cells, several thousand of short interspersed CpG-rich sequences mostly corresponding with gene promoters are devoid of methylation. Methylation patterns are shaped by the transcription factor binding to DNA and interactions between DNMTs and histone marks [4,5]. In general, the level of methylation is low to intermediate at regulatory elements and high over gene bodies and intergenic regions. If established on promoters or enhancers, methylation can repress transcription by direct inhibition of transcription factor binding or indirectly through recruitment of methyl-binding proteins and chromatin modifiers. In contrast, methylation of gene bodies is not considered a repressive mark, but may be important to prevent spurious transcription initiation [6]. A number of genomic loci evade epigenetic reprogramming in embryo development. These include the Differentially Methylated Regions (DMRs) controlling the mono-allelic and gamete of origin-dependent expression of the imprinted genes and a subset of retroviral elements [7]. Methylation at these loci is maintained through interaction with specific KRAB-zinc finger proteins [8]. 

Correct DNA methylation is required for normal human development [9]. In principle, alterations of normal methylation patterns can arise in absence of DNA sequence changes (primary epimutation) or secondary to genetic variants that can either occur in cis or in trans (secondary epimutation). Although the former can be difficult to demonstrate, there is evidence that DNA methylation can be influenced by environmental factors, especially in germ cells and during early embryogenesis [10]. Modifications of the normal DNA methylation patterns have been commonly demonstrated in cancer, for example as hypermethylation of tumor suppressor gene promoters [11]. More recently, methylated CpGs that correlate with chronological age and with risk of mortality or developing multi-factorial diseases have been demonstrated [12,13,14]. Specific DNA methylation patterns have been more rarely reported for monogenic diseases [7,9]. DNA methylation can be modified at a single locus generally as consequence of a variant occurring *in cis*, or at multiple loci possibly as consequence of a variant occurring *in trans*. In the latter case, genomic methylation landscapes may be directly altered by variants in DNMTs or indirectly by mutations in chromatin modifiers or transcription factors [9]. The recent availability of high-throughput screening platforms has led to the identifications of specific methylation signatures associated with an increasing number of disorders, indicating that DNA methylation analysis may represent a valid tool for a better classification of diseases with overlapping clinical features and for sorting cases with ambiguous genetic variants [15]. 

This review focuses on the role of DNA methylation in the molecular diagnosis of mono/oligogenic diseases. We discuss the different types of DNA methylation abnormalities, their relationship with genotype and phenotype, as well as their relevance for counselling. Furthermore, we report the main methods for detection, including in prenatal testing, and the limitations in the interpretation of results. Finally, we highlight research areas and technological advances that might extend the use of DNA methylation analysis in molecular diagnostics in the near future. The main features of DNA methylation abnormalities in human monogenic diseases are listed in Table 1.

## 2. DNA Methylation Defects and Testing 

### 2.1. Developmental Delay and/or Intellectual Disability Disorders (DD/ID)

#### 2.1.1. Fragile X Syndrome

Fragile X syndrome (FXS, OMIM#300624) is the leading inherited form of intellectual disability (ID), characterized by poor language development, hyperactivity, impulsivity, as well as other manifestations typical of an autism spectrum disorder. The estimated frequency of affected males in the general population is approximately 1 in 7000, and that of affected females 1 in 11000 [16]. The prevalence of carrier females at high risk of having an affected child is 1 in 250 or higher. 

The majority of affected individuals present an unstable mutation of >200 repeats (full mutation (FM)) of the CGG motif located at the 5′ untraslated region (UTR) of the *FMR1* gene (in Xq27.3). This structural variation arises from a smaller maternal allele with 56–200 CGG repeats known as premutation (PM) during maternal meiosis. FM (usually not PM) undergoes methylation of cytosines within the repeat itself and in the CpG island in the upstream promoter region (methylated full mutation (MFM)) [17]. The epigenetic changes (DNA methylation and accompanying histone modifications) block transcription, preventing the production of the FragileX mental retardation protein (FMRP, loss-of-function effect), although the coding sequence of the *FMR1* gene remains intact. All male carriers of methylated FM are affected by FXS, compared to only 30%–50% of carrier females. Rare male carriers of unmethylated FM alleles and with apparently normal intelligence have been described indicating that structural expansion and epigenetic changes are determined by different mechanisms [18,19,20,21]. More than 40% of FXS individuals present with either size or methylation mosaicism or both, and the percentage of methylation is negatively correlated with expression of the *FMR1* locus and the severity of ID in males [22,23]. 

The molecular diagnosis of FXS and carrier status is based on CGG repeat sizing and DNA methylation analyses. To date, the triplet repeat-primed PCR method represents a fast and precise approach to amplify expanded alleles into the FM range [24]. To analyze the methylation status of expanded alleles, a locus-specific Methylation-Specific Multiple Ligation-Dependent Probe Amplification (MS-MLPA) should be coupled to the PCR. Alternatively, triplet repeat-primed methylation-specific PCR may be used, combining allele-specific methylation PCR using a methylation-sensitive restriction enzyme and capillary electrophoresis. The latter method enables high throughput, high resolution and semiquantitative methylation assessments, as well as CGG sizing [25]. The sensitivity and specificity of these techniques are over 99%. 

Concerning recurrence risk, alleles in the normal range (<45 CGG repeats) do not involve a risk to offspring, despite positive family history. PM alleles are highly unstable during maternal transmission and tend to expand to FM in one generation. Expansions may also occur in paternal transmission, although remain within the PM range. All daughters of a PM male are obligatory PM carriers. Likewise, all daughters of rare unmethylated FM individuals are obligate carriers, due to inheritance of a PM allele. On the other hand, rare cases of methylated PM have the same recurrence risk of PM carriers. All FXS children have a PM or FM carrier mother, who has a 50% risk of passing on the expanded allele. 

#### 2.1.2. Chromatin-Related Disorders

Chromatin-Related Disorders (CRDs) are caused by genetic alterations of components of the epigenetic machinery and represent 5%–10% of developmental disorders [155]. These defects alter the balance of proteins controlling the expression of many genes by modifying DNA, histones and chromatin structure. According to the literature, 82 human conditions associated with mutations in 70 epigenetic machinery genes are described [156]. Although these diseases may be distinguished by ancillary features and characteristic facial dysmorphisms, clinical and molecular overlap (including intellectual disability, growth retardation and immune dysfunction) likely resulting from convergent pathways is often reported.

The molecular diagnosis of CRDs is currently based on exome- and/or gene-targeted sequencing. Recently, blood-derived DNA methylation signatures (epi-signatures) have been identified as highly specific marks in an increasing number of CRDs, regardless of the primary chromatin alteration (i.e., histone methylation, histone acetylation, etc.) [15]. These epi-signatures are detected by interrogating whole-genome methylation arrays, such as the Illumina Infinium MethylationEPIC BeadChIP, and have proved to be a valid tool for distinguishing affected and unaffected individuals, as well as pathogenic and non-pathogenic variants [15]. The first example of the clinical utility of whole-genome epi-signatures has been reported for Claes–Jensen syndrome (CJS), an intellectual disability caused by mutations in the X-linked *KDM5C* gene, encoding a histone H3 lysine 4 (H3K4me3) demethylase [26]. The use of a more extended methylation array has recently allowed researchers to demonstrate 1769 differentially methylated CpGs, mostly within nine genomic regions in the peripheral blood of CJS-affected males and (with intermediate level) carrier females [27]. Consistent with the inverse correlation between DNA methylation and H3K4me3, most of these CpGs overlapping protein-coding genes are located close to CpG islands and are hypomethylated in patients relative to controls [28].

An even more striking example of extensive methylation defect is provided by Sotos syndrome (SS), which is caused by mutations in the histone H3K36 methyltransferase *NSD1* gene [30]. In this case, >7000 CpG were found hypomethylated in patients versus controls, consistent with the demonstrated interaction between H3K36me3 and de novo DNMTs [30]. Significantly, blood SS epi-signatures allowed researchers to discriminate between pathogenic *NSD1* mutations and benign *NSD1* variants [30]. As part of their methylation defect, an acceleration of the Horvath’s epigenetic aging clock was also demonstrated in SS patients [31].

Another epi-signature detectable in peripheral blood was reported for Kabuki syndrome (KS) [32]. In this case, a similar methylation pattern was observed for both Type 1 (KS1, OMIM #147920) and type 2 (KS2, OMIM #300867) Kabuki syndrome variants, which are caused by mutations in the histone methyltransferase *KMT2D* and histone demethylase *KDM6A* genes, respectively. The methylation disturbance is complex, with 856 hypomethylated and 648 hypermethylated CpGs, and the most differentially methylated regions overlap with protein-coding genes and CpG islands, including the *MYO1F* gene and the *HOXA5* and *HOXA-AS3* promoters. This epi-signature allowed researchers to distinguish KS resulting from *KMT2D* loss of function mutations from KS-like phenotypes with other etiologies. Moreover, the comparison of the KS-specific epi-signature with that of healthy controls allowed researchers to re-classify variants of unknown significance (VUS) into benign and likely pathogenic variants. A similar result has been obtained for CHARGE syndrome (coloboma, heart anomaly, choanal atresia, retardation, genital and ear anomalies, OMIM #214800), whose specific epi-signature has been used for classifying the pathogenicity of VUS in the chromodomain-containing helicase *CHD7* gene [33]. Interestingly, methylation patterns also allowed researchers to discriminate between CHARGE and Kabuki cases with extensive clinical overlap. In particular, *HOXA5* is hypermethylated in both CHARGE and Kabuki-specific signatures and may account, together with a few other targets, for some of the clinical overlap between these disorders, but the majority of CpGs of the CHARGE epi-signature are specific and located within genes related to neural growth and development and other relevant functions. 

Methylation abnormalities in 16 regions across the genome constitute the highly specific epi-signature in the peripheral blood of patients with alpha thalassemia/mental retardation X-linked (ATRX) syndrome (OMIM #301040), which is caused by alterations in the chromatin remodeling factor ATRX [34]. This protein is involved in chromosome segregation, DNA repair, and transcriptional regulation. Moreover, ATRX is involved in the control of DNA methylation at subtelomeric and repetitive regions and has been shown to interact with the methyl-binding protein MECP2. In patients with ATRX syndrome, the majority of the CpGs of its epi-signature are hypermethylated. These abnormally methylated CpGs are not restricted to telomeric and pericentromeric regions, but include promoter-associated CpG islands and protein-coding genes, such as those of the transcriptional regulators PRDM9, ZNF274 and ZNF300, possibly contributing to the ATRX syndrome phenotype. 

Another example of a highly specific epi-signature is provided by Floating–Harbor syndrome (FHS, OMIM #136140), which is associated with heterozygous variants in the chromatin remodeling *SRCAP* gene. A total of 28 hyper or hypomethylated regions preferentially occurring in CpG islands, including those associated with the *FIGN* and *STPG2* (hypermethylated), and *MYO1F* and *RASIP1* (hypomethylated) genes, have been identified in these patients [35]. Specific epi-signatures have also been demonstrated in patients carrying variants in different subunits of the BRG1-associated factors (BAF) chromatin remodeling complex (disorders also known as BAFopathies). The finding of similar methylation patterns (e.g., consistently hypermethylated regions overlapping the keratin KRT8 and KRT18 genes) in Coffin–Siris syndrome (OMIM #135900, 614608, 614609), Nicolaides–Baraitser syndrome (OMIM #601358) and chromosome 6q25 deletion (OMIM #612863) cases confirms the functional link within this group of disorders [36]. Moreover, in these cases, the identified epi-signatures have proven to be able to resolve ambiguous clinical cases, as well as to re-classify VUS. 

A different scenario was demonstrated for Activity-dependent neuroprotector homeobox (ADNP) syndrome (also known as Helsmoortel–Van der Aa syndrome, OMIM #615873). This disorder is associated with dominant negative truncating variants in the neuroprotective transcription factor ADNP gene. However, two distinct epi-signatures have been found in patients affected by this disease, suggesting its possible reclassification into two separate entities [37]. In particular, hypomethylation of ~6000 CpGs is associated with mutations in the amino-terminal half, while hypermethylation of ~1000 CpGs is associated with mutations involving the central nuclear localization signal of the ADNP protein [37]. The two epi-signatures are only partially overlapping and both involve genes mostly related to neuronal functions. These methylation patterns have proven to be able to identify cases of ADNP syndrome within cohorts of patients with unresolved developmental delay (DD)/ID.

Specific epi-signatures are also associated with *DNMT* variants. In particular, patients affected by Autosomal Dominant Cerebellar Ataxia with Deafness and Narcolepsy (ADCA-DN, OMIM #126375), which is caused by dominant missense variants in the Replication Foci Targeting Sequence of DNMT1, show both moderate global DNA hypomethylation, and hypermethylation of specific loci, including gene bodies, intergenic regions, promoters and CpG islands [29,38]. Hereditary Sensory Neuropathy with Dementia and Hearing Loss (HSAN1E, OMIM #614116) is caused by mutations in different residues of the same domain of DNMT1, and has strong clinical overlap with ADCA-DN. Patients affected by HSAN1E also have moderate global hypomethylation and specific hypermethylation that have both been linked with neurological disease, but it is still unclear how similar or different the epi-signatures of these two disorders are [39]. Opposite methylation changes have been found in Tatton-Brown–Rahman syndrome (TBRS, OMIM #615879) and Heyn–Sproul–Jackson syndrome (HESJAS, OMIM #618724), which are associated with loss of function and gain of function variants in the *DNMT3A* gene, respectively. Loss of methylation of 388 regions corresponding to intergenic regions and CpG island shores significantly enriched at genes involved in development and growth pathways, as well as an acceleration of the Horvath’s epigenetic aging clock, have been described in TBRS patients [40]. Conversely, hypermethylation in the majority of deregulated regions, including evolutionary conserved regions associated with Polycomb-regulated developmental genes, were found in HESJAS cases [41]. Concerning Immunodeficiency with Centromeric Instability and Facial Anomalies syndrome (ICF), different epi-signatures have been demonstrated in blood DNA, in the four molecular subgroups of this disorder [42]. Patients affected by ICF1 (OMIM #242860), in which the genetic defects are represented by recessive loss of function mutations in *DNMT3B*, show hypomethylation of pericentromeric repeats (satellites two and three), subtelomeric regions, other repetitive elements, and CpG island-associated promoters of germline-specific genes. The epi-signatures of ICF2 (OMIM #614069), ICF3 (OMIM #616910) and ICF4 (OMIM #616911) that are associated with recessive variants of the transcription factors/chromatin modifiers ZBTB24, CDCA7 and HELLS, respectively, share with the ICF1 epi-signature hypomethylation of pericentromeric repeats, but are characterized by further specific hypomethylation of CpG-poor genomic regions with hallmarks of heterochromatin, including gene clusters expressed in a random or imprinted monoallelic manner.

Specific epi-signatures have been recently demonstrated in other CRDs, including Genitopatellar and Say–Barber–Biesecker–Young–Simpson syndromes, Werner syndrome, Williams and 7q11.23 duplication syndromes, progressive supranuclear palsy and frontotemporal dementia, Cornelia de Lange and SETD1B-related syndromes [15,29,44,45]. Details on the aetiology and methylation patterns of these disorders are reported in Table 1. Overall, limited overlap among the epi-signatures, mainly involving genes related to histone modifications (*PRDM9*, *HIST1H3E*, *NSD1* and *SETDB1*), has been demonstrated, which should allow concurrent classification of all CRDs through genome-wide mCpG analysis. As an example, only 217 out of 15,408 CpGs included in the epi-signatures of nine CRDs are shared by more than two conditions, and only 18 CpGs by more than three [29]. Blood methylation profiles are influenced by cell-type composition [157]. However, at least in a cohort of TBRS patients, the observed epi-signatures were not significantly influenced by cell-type variation [40]. Moreover, for some CRDs, reproducible epi-signatures have been obtained in both peripheral blood leukocytes and fibroblasts, after filtering out tissue-specific differences [41,43].

### 2.2. Imprinting Disorders

Imprinting Disorders (ImpDis) are a clinically heterogenous group of diseases, of which the common feature is dysregulation of imprinted genes [95]. Overall, ImpDis can be considered the prototype of disease for which the DNA methylation pattern that is detected in blood leukocytes is sufficient for diagnosis [95]. They generally present a methylation abnormality in the locus that is directly responsible for the clinical phenotype. Thus, molecular diagnosis is based on the finding of loss or gain of methylation (LOM or GOM) of the germline-derived DMR regulating the imprinting of the locus (also known as the Imprinting Centre (IC)). However, a subgroup of patients often showing more complex phenotypes have Multi-Locus Imprinting Disturbances (MLIDs) [96]. Imprinted methylation abnormalities are often associated in cis with genetic defects (e.g., Copy-Number variants (CNVs), Uniparental Disomy (UPD) or Single Nucleotide Variants (SNVs)), but can also occur in the absence of obvious genetic change as primary epimutations. MLIDs have been associated with genetic variants occurring in trans, either in the zygote or maternal oocyte [7]. Molecular diagnosis of ImpDis is commonly obtained with methylation analysis of the germline-derived DMR of the associated locus [95]. Currently, the most commonly used approach to reach the molecular diagnosis is MS-MLPA on peripheral blood DNA, allowing simultaneous detection of DNA methylation and CNVs. A specific MS-MLPA assay for each ImpDis and a further one targeting to multiple loci for detecting MLID are commercially available [97]. Recently, the use of a genome-wide methylation assay with methylation arrays has been implemented and proven to be particularly useful for detection of MLID [98,99,100]. Methylation abnormalities occurring in individual ImpDis and in MLID will be treated separately.

#### 2.2.1. Prader–Willi Syndrome and Angelman Syndrome

Prader–Willi syndrome (PWS; OMIM #176270) and Angelman syndrome (AS, OMIM #105830) are neurodevelopmental ImpDis caused by different genetic and epigenetic defects in the chromosome 15q11–q13 region. Their prevalence at birth is 1:10,000–1:25,000 for PWS and 1:12,000–1:20,000 for AS. The locus harbors the imprinted genes *UEB3A* and *ATP10C* that are expressed from the maternal allele, and a group of paternally expressed genes including five protein-coding genes (*MKRN3, MAGEL2, NDN, C15orf2* and the bicistronic *SNURF-SNRPN*), a cluster of small-nucleolar RNA (snoRNA) genes including *SNORD116*, and several antisense transcripts (including the antisense transcript to *UBE3A* or *SNHG14*). Imprinting of the locus is controlled by a bipartite IC consisting of the PWS-IC and the AS-IC. The PWS-IC overlaps the *SNURF*:Transcription Start Site (TSS)-DMR and directs gene expression from the paternal chromosome, while the AS-IC corresponds to an oocyte-specific promoter that is necessary for de novo methylation of the PWS-IC in the maternal germline [46]. Some of the paternally expressed genes (e.g., *MAGEL2*) have their promoter marked by a somatic maternally methylated DMR that is hierarchically controlled by the *SNURF*:TSS- DMR. Loss of function or expression of the paternally expressed genes causes PWS, while loss of function or expression of the maternally expressed *UBE3A* results in AS [47].

PWS is characterized by muscular hypotonia, intellectual disability, short stature, hyperphagia-driven obesity, hypogonadotropic hypogonadism and small hands and feet [48]. The exact contribution of each gene of the 15q11-q13 cluster to the PWS phenotype is unclear. However, the minimal region lost in some patients carrying small atypical deletions harbors *SNORD 116*, which is therefore considered the major gene contributing to the PWS phenotype. The most common molecular defects are: 5–7 Mb de novo deletions, removing the entire imprinting cluster on the paternal chromosome (accounting for up 70%–75% of patients), maternal UPD of chromosome 15 (found in 25% of patients), epigenetic silencing of the paternal allele caused by an imprinting defect, switching the paternal to maternal imprints (i.e., gain of methylation of the paternal *SNURF*:TSS- DMR allele) in 1% of the cases. The imprinting defect can be due to microdeletions of the IC or primary epimutations, occurring without any detectable change in the DNA sequence in cis. In less than 1% of the patients carrying paternal deletions, the defect can result from a parental balanced chromosome 15 rearrangement [48].

Over 99% of PWS cases can be diagnosed analyzing DNA methylation and CNVs at 15q11–q13 by MS-MLPA. In all cases, gains in methylation of the *SNURF*:TSS-DMR and somatic 15q11–q13 DMRs can be detected in blood DNA [49]. Microsatellite analysis should be performed to distinguish maternal UPD 15 from primary IC epimutation. The recurrence risk is dependent on the genetic mechanisms underlying the PWS. Low recurrence risk is reported for patients with de novo deletions, maternal UPD 15 and primary imprinting defect in absence of genetic mutation. Recurrence risk is higher in the presence of inherited chromosome 15 rearrangements, and corresponds to 50% in imprinting defect cases with paternal inheritance of IC microdeletions [50]. Early diagnosis, preferably in the nursery, offers the opportunity to greatly improve the health and quality of life of the patients and their families, as management of PWS is very age-dependent and focused on anticipatory guidance and addressing the consequences of the syndrome. 

AS is a neurogenetic disorder, characterized by microcephaly, severe intellectual deficit, speech impairment, epilepsy, electroencephalogram abnormalities, ataxic movements, tongue protrusion, paroxysms of laughter, abnormal sleep patterns, and hyperactivity [51]. The molecular defects are similar to those causing PWS, but affecting the maternal chromosome. Typical 5–7 Mb de novo deletions of 15q11.2–q13 on the maternal chromosome are found in 75% of patients, paternal UPD of chromosome 15 is found in 1%–2% [51], and imprinting defects (i.e., loss of methylation of the maternal *SNURF*:TSS-DMR) resulting from IC microdeletion or mosaic primary epimutation is found in 3% [49,52,53]. Furthermore, 5%–10% of AS patients have normal methylation, but present clinical variants in the maternal allele of *UBE3A*; 10%–15% are idiopathic.

Consensus diagnostic clinical criteria for AS have been developed [54,55], but clinical diagnosis in infants and young children is sometimes difficult, as the unique clinical features of AS may not manifest until after one year of age, so confirmation by genetic testing can be very useful; it is also useful for determining the risk of recurrence. A methylation test of 15q11–q13 is required in patients with ID if the following clinical features are also present: normal or high birth weight, developmental delay and obesity with food-seeking behavior. As with PWS, for AS the molecular diagnosis is performed by MS-MLPA and microsatellite analysis of 15q11–q13. Targeted sequencing of *UBE3A* is required in case of a negative methylation test. The recurrence risk is low in the case of a de novo deletion, UPD and imprinting defect without IC-deletion. In case of *UBE3A* variants and imprinting defects caused by IC microdeletions, the recurrence risk is 50% if the defect is inherited from the mother, low if de novo arises and higher in the presence of maternal germline mosaicism [47].

#### 2.2.2. Temple Syndrome and Kagami–Ogata Syndrome

Temple syndrome (TS; OMIM #616222) and Kagami–Ogata syndrome (KOS; OMIM #608149) are very rare ImpDis caused by imprinting alterations of the 14q32.2 region. This region harbors several maternally expressed noncoding RNA genes (*MEG3*, *RTL1as* and many small nucleolar and micro RNAs, likely as a single polycistronic transcript), as well as paternally expressed genes (*DLK1* and *RTL1*), whose imprinting is regulated by the paternally methylated *MEG3/DLK1*:Intergenic (IG)- DMR. The somatic *MEG3*:TSS–DMR and *MEG8*:Int2–DMR are methylated in the paternal and maternal chromosome, respectively, and their methylation is hierarchically controlled by the *MEG3/DLK1*:IG-DMR. 

TS is characterized by early-onset hypotonia, feeding difficulties, short stature, precocious puberty, obesity and brachydactyly. About 20% to 50% of patients have phenotypic overlap with Silver–Russell and Prader–Willi syndromes, especially in early childhood. 

The most common molecular mechanisms in TS are maternal UPD 14 (accounting for up to 78% of TS patients), paternal microdeletions of 14q32.2 (9.8%), and primary epimutations (hypomethylation) of the *MEG3/DLK1*:IG-DMR (ranging from 11% to 60%) [56,57]. These lesions lead to overexpression of the paternally expressed genes and expression loss of *DLK1* [58]. Recently, intrachromosomal triplications with runs of homozygosity (rare, postzygotic events, with undetectable mosaicism rate) have been reported as a potential mechanism causing segmental uniparental disomy in TS [59]. Mosaicisms occur in about 50% of cases. The diagnosis is reached using MS-MLPA and demonstrated as hypomethylation of the *MEG3/DLK1*:IG-DMR and *MEG3*:TSS–DMR. MLIDs have been reported in TS but not in KOS patients [60]. Recurrence risk depends on the molecular mechanism; low recurrence risk is present in case of UPD unless an inherited rearrangement of chromosome 14 is present, in case of microdeletions unless a balanced rearrangement or germinal mosaicism is present in the mother, and in case of primary methylation defects. KOS is a severe, extremely rare condition characterized by peculiar facial features, polyhydramnios and omphalocele, abnormality of the costal arch, bell-shaped thorax and early lethality. The molecular defects in patients with KOS result in lack of expression of the maternally expressed genes and overexpression of *DLK1*. The diagnosis can be obtained by MS-MLPA demonstrating hypermethylation of the *MEG3/DLK1*:IG-DMR and *MEG3*:TSS–DMR [61]. The most common molecular mechanisms are UPD14 pat (about 60%), maternal microdeletions of 14q32.2 of variable size, but mostly including the *MEG3/DLK1*:IG-DMR and/or *MEG3*:TSS–DMR (20%) (rare microdeletions involve the maternally expressed noncoding RNAs [62]) and primary epimutations (hypermethylation) of the *DLK1/MEG3:IG-DMR* (20%). One single report describes mosaic pat UPD14 [63]. Genetic counselling for recurrence risk is based on the molecular mechanisms. UPD, unless related to an inherited rearrangement of chromosome 14, has low recurrence risk; microdeletions, if a balanced rearrangement is not present in the mother or a germinal mosaicism exists, have a low recurrence risk; primary DMR methylation defects have a low recurrence risk. 

#### 2.2.3. Beckwith–Wiedemann Syndrome and Silver–Russell Syndrome

Beckwith–Wiedemann syndrome (BWS, OMIM #130650; prevalence at birth: 1:10500) and Silver–Russell syndrome (SRS; OMIM #180860; prevalence at birth 1:30,000/1:100,000) are different ImpDis, both associated with the imprinted genes of chromosome 11p15. BWS is characterized by variable clinical features, which may include macroglossia, abdominal wall defects, overgrowth, lateralized overgrowth, organomegalia and predisposition to embryonal tumors [64]. A recent international consensus document has recognized the existence of a Beckwith–Wiedemann spectrum (BWSp) covering classical BWS without a molecular diagnosis and BWS-related phenotypes with an 11p15.5 molecular anomaly [64]. The typical BWS molecular abnormalities affect one or both of two functional domains of the imprinted gene cluster located at chromosome 11p15.5–11p15.4 [65]. The telomeric domain harbors the insulin-like growth factor 2 (*IGF2*) gene that is expressed from the paternal allele and encodes a protein promoting fetal growth, and *H19*, which is expressed from the maternal allele and encodes a non-translated long non-coding RNA with growth inhibitory properties. Their reciprocal imprinting is regulated by the *H19/IGF2*:IG-DMR (also known as IC1), whose unmethylated maternal allele acts as a CTCF-binding-dependent insulator. The centromeric domain harbors a group of genes expressed from the maternal chromosome, including the growth inhibitor *CDKN1C*. These genes are repressed on the paternal chromosome by the long non-coding RNA *KCNQ1OT1*, whose promoter is maternally methylated and overlaps the *KCNQ1OT1*:TSS-DMR (also known as IC2).

A molecular defect affecting imprinted genes in the chromosome region 11p15 can be detected in ~85% of patients [64,65]. DNA methylation changes are the most frequent abnormalities. IC2 LOM leading to *KCNQ1OT1* upregulation and *CDKN1C* repression in the maternal chromosome can be found in ~50% of cases. About one third of these patients shows MLID (see below). 5%–10% of BWS patients have IC1 GOM that results in increased *IGF2* and reduced *H19* expression on the maternal chromosome. In about one third of these cases, IC1 LOM is associated with maternal 1.4–2.2 Kb deletions or single nucleotide variants (SNV) inside IC1, which are believed to be predisposed to the methylation defect (secondary epimutations). Both IC1 GOM and IC2 LOM resulting from mosaic segmental paternal 11p15 UPD (upd(11)pat) can be found in 20% of patients. Up to 10% of these have mosaic genome-wide UPD (paternal unidiploidy). IC1 GOM or IC2 LOM, or both, can also be detected in BWS patients with 11p15 chromosome abnormalities, which account for about 1% of cases and correspond to more frequent paternal duplications and rare maternal deletions. About 5% of BWS cases have normal 11p15 methylation and carry intragenic loss of function maternal variants in *CDKN1C*. 

Molecular diagnosis of BWS is obtained by detection of abnormal methylation of IC1, IC2 or both, and is most commonly reached by using MS-MLPA, which also recognizes CNVs [64,65]. A further microsatellite analysis is needed to confirm upd(11)pat, and Sanger sequencing is needed to detect possible clinical variants in IC1 and *CDKN1C*. A total of 15% idiopathic cases can be due to low-level mosaicism or other undefined mechanisms. Indeed, mosaicism is a major challenge for molecular diagnosis of BWS and SRS [66]. Modest methylation abnormalities may not be detected, because of the limited sensitivity of the method used, leading to false negative results. In some BWS patients, differences in IC1 and IC2 methylation have been observed between blood and tongue, and in one case even between the two sides of the tongue [67,68]. Thus, in case of unequivocal clinical diagnosis and negative molecular testing in blood, analysis of DNA derived from other tissues (e.g., fibroblasts, oral mucosa) might be considered [64]. 

The recurrence risk for BWS is generally low in case of upd(11)pat and IC1 and IC2 epimutation. However, the cases with IC1 GOM and carrying IC1 deletion/SNV have a 50% risk of transmitting the disease via maternal transmission. In addition, increased recurrence risk has been reported in cases with MLID with maternal-effect clinical SNVs (see ref [113] and MLID section). Moreover, high recurrence risk via maternal transmission has been described in rare cases of IC2 LOM with *KCNQ1OT1* rearrangements or clinical SNVs [69]. Furthermore, a 50% recurrence risk is also present in the cases with clinical *CDKN1C* SNVs (only via maternal transmission), and in the cases of inherited chromosomal rearrangements (parental bias depends on the type of mutation). 

SRS is a congenital developmental disorder characterized by pre- and post-natal growth retardation, craniofacial features (triangular shaped face and broad forehead), relative macrocephaly at birth, body asymmetry and feeding difficulties. Although clinical scoring systems have been proposed by several groups [70,71,72], the accuracy of the diagnosis is often difficult in less severely affected individuals, because of clinical heterogeneity and attenuation of clinical features with aging.

SRS can be regarded as the genetically (and clinically) opposite disease to BWS, as demonstrated by specular molecular defects at chromosome 11p15 [73,74]. The primary molecular cause of SRS is IC1 LOM, which leads to reduced *IGF2* and increased *H19* expression and accounts for 40%–60% of patients [70]. In a subgroup of cases, IC1 LOM is present in mosaic form and its detection is challenging, because tissues other than blood may be more severely affected [75]. In rare cases, IC1 LOM is associated with in cis microdeletions within the paternal IC1 [76]. In a significant proportion (15%–38%) of IC1 LOM cases, the methylation defect affects also other DMRs (see MLID section) [77]. Duplications (generally of maternal origin) and deletions affecting the 11p15.5 imprinting cluster account for 1%–2% of patients [78]. Single familial cases with gain of function *CDKN1C* or loss of function *IGF2* variants and normal methylation have also been reported [79,80]. The locus 11p15.5 is not the only imprinting locus to be involved in SRS—total or segmental maternal UPD of chromosome seven (upd(7)mat) is found in 5%–10% of patients [81,82]. The critical region of this chromosome contributing to the SRS phenotype has not been established yet, although there are currently three candidate imprinted loci (*MEST/PEG1*, *GRB10* and *PEG10*) in which an isolated epigenetic change may conceivably lead to SRS phenotype. Patients with SRS-like phenotype may have abnormalities affecting the 14q32 imprinted gene cluster, consistent with clinical overlap between SRS and TS [56]. The molecular aetiology of SRS remains unknown in about 30%–40% of patients. 

Diagnostic genetic testing of SRS can be performed by detecting methylation disturbances and CNVs in IC1 of chromosome 11p15, and *MEST*:alt-TSS-DMR and *GRB1*:alt-TSS-DMR of chromosome seven, by MS-MLPA. Microsatellite analysis is needed to confirm upd(7)mat and upd(11p15)mat. The recurrence risk for SRS is low in case of upd(7)mat and primary IC1 epimutation, but it is 50% with parental bias in cases of inherited chromosomal rearrangements/clinical variants/IC1 microdeletions, and is increased in cases of MLID with maternal clinical variants (see MLID section). 

#### 2.2.4. Pseudohypoparathyroidism

Pseudohypoparathyroidism (PHP) represents a group of disorders characterized by resistance to the parathyroid hormone (PTH) leading to hypocalcemia and hyperphosphataemia [83]. PHP is also defined by several clinical features such as brachydactyly, short stature, stocky build, obesity, round face and subcutaneuous ossification, also known as Albright hereditary osteodsystrophy (AHO). In 50%–75% of the cases, cognitive impairment is also present. Pseudopseudohypoparathyroidism (PPHP, OMIM #612463) is characterized by AHO, without hormone resistance [84]. PHP and PPHP are primarily caused by a molecular defect of the *GNAS1* locus that encodes the G protein (G_s_-alpha) involved in the signaling pathway of PTH and other hormones, through the activation of cAMP [85]. The *GNAS1* locus is located in chromosome 20q in a region regulated by imprinting [86]. Most of the tissues show biallelic *GNAS1* expression, while predominant maternal expression is present in endocrine tissues, such as the thyroid, ovary and pituitary gland. 

Based on clinical features, PTH hormone resistance, cAMP response and protein G_s_-alpha activity, PHP is classified in PHP Ia (OMIM #103580), PHP Ib (OMIM #603233), PHP Ic (OMIM #612462) and PPHP [87]. PHP Ia shows AHO features and, usually, multiple hormone resistance, decreased cellular cAMP response to PTH infusion and decreased erythrocyte protein G_s_-alpha activity [88]. PHP Ib has renal PTH resistance, a decreased cAMP response to PTH infusion, normal erythrocyte G_s_-alpha activity and patients do not show signs of AHO. PHP Ic is characterized by PTH resistance, generalized hormone resistance, AHO, decreased cAMP response to PTH infusion, and normal erythrocyte G_s_-alpha activity. PPHP shows AHO without endocrine abnormalities, a normal cellular cAMP response to PTH infusion and decreased erythrocyte G_s_-alpha activity.

In 70%–80% of the cases, PHP Ia is caused by maternally inherited inactivating mutations of the *GNAS1* gene [89]. In a few cases, large deletions, including part of or the whole gene, have been reported. Only PHP Ib is characterized by methylation abnormalities; in all these cases, at least one among the three maternally methylated DMRs and the somatic paternally methylated DMRs of the locus is affected [90]. In the familial form (15%–20% of the cases), which is associated with maternal microdeletions affecting the *STX16* gene or the *GNAS* locus, the defect may be limited to the LOM of the *GNAS A/B*:TSS-DMR (in the case of STX deletion) or be extended to the other DMRs (in cases with *GNAS* locus deletion). In most of the sporadic cases, more than one DMR is involved. In 8%–10% of the PHP 1b cases, the methylation abnormality is caused by maternal UPD of chromosome 20 (upd(20q)mat). MLID may be present in PHP 1b patients, although evidence is limited. PHP Ic patients have a maternal inactivating mutation in the C-terminal portion of GNAS1 that is required for receptor-mediated activation, while PPHP is caused by paternally inherited inactivating mutations (point mutations or large deletions) of the *GNAS1* locus [91]. 

Molecular diagnosis of PHP 1b is usually done by MS-MLPA to identify methylation defects and deletions encompassing the *GNAS1* locus. Microsatellite analysis is required to identify upd(20q)mat. Recurrence risk for PHP 1b is low, unless a maternally inherited deletion in the *STX16* or *GNAS* loci is present.

#### 2.2.5. Transient Neonatal Diabetes Mellitus

Transient neonatal diabetes mellitus (TNDM, OMIM#601410; prevalence at birth: 1:400,000) is defined as an insulin-requiring hyperglycemia appearing during the first six months of life. In about half of the neonates, diabetes is transient and resolves before five months. In a significant number of patients, type II diabetes appears later in life. Some patients present low birth weight and congenital abnormalities, such as macroglossia and umbilical hernia, while, less frequently, dysmorphic facial features, renal tract anomalies and cardiac anomalies are present [92]. 

The major cause of TNDM (70% of the cases) is overexpression of the imprinted genes *PLAGL1* and *HYMA1* on chromosome 6q24 [93]. This gene dysregulation can be caused by (i) paternal UPD of chromosome 6 (UPD(6)pat), (ii) paternally inherited duplication of 6q24 or (iii) maternal LOM of the *PLAGL1*:alt-TSS-DMR that controls the imprinting of the locus. In about 40% of the patients with LOM, the methylation defect is isolated, while the remaining cases have MLID. Around 50% of these patients carry loss-of-function recessive mutations of *ZFP57,* a transcription factor required for the maintenance of methylation during early embryonic development [94]. Molecular diagnosis is usually done by MS-MLPA to identify UPD(6)pat, paternal 6q24 duplications and single or multi-locus LOM. Microsatellite analysis is needed to distinguish cases of paternal UPD from maternal LOM. Sanger sequencing is needed to identify clinical SNVs in the *ZFP57* gene. Recurrence risk is low in cases of UPD(6)pat and single-locus LOM, it is 50% in cases with paternally inherited duplications, and 25% in MLID cases with clinically significant *ZFP57* SNVs.

#### 2.2.6. Multilocus Imprinting Disturbances

Epimutations associated with ImpDis may affect not only single but also multiple imprinted loci, which may become abnormally methylated throughout the genome. This phenomenon has been defined as a Multilocus Imprinting Disturbance (MLID) [7]. The first reported evidence is represented by a few cases of TNDM, which, in addition to 6q24 LOM, also displayed LOM at the 11p15 IC2 and led the researchers to hypothesize that the TNDM and BWS loci were functionally related [101,102]. Since then, a series of targeted and genome-wide studies revealed that MLID is a recurrent event in most ImpDis. As for the epimutations limited to a single locus, for MLID, intra- and inter-tissue mosaicism is observed [103]. The reported frequency of MLID varies among ImpDis and appears quite heterogeneous in cohorts with the same disorder, probably biased by the use of different and targeted rather than genome-wide approaches. The highest frequency (50%–75%) has been reported in TNDM patients with *PLAGL1:alt*-TSS–DMR hypomethylation [104,105]. In BWS patients with IC2 LOM, the incidence ranges between 20% and 50% [104,105,106,107,108,109], in SRS patients with IC1 LOM between 9.5% and 30% [71,104,105,107,110], and in PHP-1b patients with GNAS imprinting defects from 0% to 6.3% [111,112]. MLID is rare in AS and PWS [105].

Although several targeted methods have been set up for the molecular diagnosis of MLID differing in its sensitivity and the number of DMRs analyzed, the MRC–Holland Multilocus MS-MLPA probably represents the best approach for a preliminary, fast and cheap screening. Genome-wide approaches based on the HumanMethylation 450 K and Infinium Methylation EPIC 850 K platforms allow us to investigate a higher number of imprinted DMRs and may increase the number of ImpDis cases scoring positive for MLID [100,113]. When genomic DNA from both blood leucocytes and oral mucosa were processed, some discrepancies in either the degree of methylation or number of affected DMRs were observed, consistent with the mosaic nature of the phenomenon [100]. 

Epigenotype-phenotype correlation studies of MLID cases are limited and the clinical relevance of this phenomenon is still unclear. Indeed, the majority of the patients with MLID show clinical features that are characteristic of one ImpDis (more frequently, BWS, SRS or TNDM). However, in some cases, phenotypic differences have been described between the cohort with MLID and that with isolated methylation defects, suggesting that some imprinted loci may act as modifier genes [114,115].

Concerning the causal mechanisms of MLID, pathogenic variants in maternal-effect genes mostly coding for members (*NLRP5*, *NLRP2*, *NLRP7* and *PADI6*) of the subcortical maternal complex have been found in women with reproductive problems and their offspring, including individuals with characteristic features of ImpDis (mostly BWS or SRS) and developmental delay or behavioral problems, and recurrent miscarriages [100,108,116,117,118]. Moreover, in patients affected by TNDM and with a specific pattern of MLID, recessive loss of function variants have been found in the *ZFP57* gene, which is a zinc finger protein interacting with the methylated allele of germline-derived DMRs [158,159].

### 2.3. DNA Methylation Defects in Hereditary Tumours

While somatically acquired epigenetic changes are common in tumor cells, a small fraction of patients harbor a constitutional epimutation predisposed to cancer. This mechanism is well documented for retinoblastoma and Lynch syndrome. Constitutive methylation has been described for other cancer predisposing and tumor suppressor genes, including *BRCA1* and *DAPK1* [160,161]. However, the frequency and relevance of this mechanism for tumor predisposition has yet to be determined, and currently it is not recommended to investigate epimutations in these genes in a diagnostic setting. 

#### 2.3.1. Retinoblastoma 

Retinoblastoma (OMIM #180200), representing the most frequent ocular cancer of the pediatric age, is caused by biallelic inactivation of *RB1* tumor suppressor gene [119]. Although it is well known that methylation of the *RB1* promoter is a rather frequent inactivating event in retinoblastoma cancer cells, the role of constitutive epimutations has been poorly investigated and it has long been controversial [120,121,122]. However, a recent study demonstrated that *RB1* promoter methylation can act as the first ‘hit’ in rare cases of retinoblastoma [123]. The presence of mosaic promoter methylation segregating with the disease was demonstrated by MS-MLPA in the blood of one out of 50 retinoblastoma patients who tested negative for germline predisposing variants. Methylation affected the maternal allele, which is normally preferentially expressed [124], and had a strong impact on *RB1* expression. Bisulfite pyrosequencing confirmed the aberration in DNA isolated from oral mucosa, although at lower levels (mean ~34% vs. 49% in blood). The probability of transgenerational inheritance is very low, but these data suggest that surveillance for the onset of second primary malignancies is appropriate in such cases. 

#### 2.3.2. Lynch Syndrome 

Lynch syndrome (LS) is caused by alterations in the mismatch repair (MMR) genes *MLH1*, *MSH2*, *MSH6* and *PMS2*. Tumors arising in LS subjects usually display absent or reduced immunohistochemical expression of the protein corresponding to the gene that is altered in the germline. In addition, about 5%–15% of colorectal carcinomas (CRCs) display reduced MLH1 immunostaining and MMR deficiency due to promoter hypermethylation. In most cases, epigenetic inactivation is a somatic event, but constitutional *MLH1* hypermethylation has been reported in several patients [125,126,127,128]. Overall, this mechanism explains 1.5%–10.5% of CRCs associated with abnormal expression of *MLH1*, and might account for up to 3% of LS [128]. In a few cases, the epigenetic abnormality is accompanied by deletions or single nucleotide substitutions, namely a specific *MLH1* haplotype bearing the variants c.27C>A and c.85G>T (secondary epimutation), and inheritance of these genetic variations and associated epimutation has been observed in some families [129]. However, more frequently, the aberrant methylation pattern is not accompanied by DNA sequence alterations (primary epimutation) and is usually reverted across generations, although occasional inheritance has been observed [130]. *MLH1* epimutations tend to present as sporadic cases affected with multiple primary tumors, with an earlier average age at cancer diagnosis compared to the general population [128]. A handful of mosaic cases have been reported, including a case showing levels of methylated *MLH1* alleles as low as 1% in blood and other tissues, which implies an opportunity to use very sensitive techniques [131]. The most commonly used method for *MLH1* promoter methylation analysis is MS-MLPA, usually performed on peripheral leukocytes [135]. However, while this method is valuable for screening, it is semiquantitative. Therefore, additional approaches should be used, including pyrosequencing and MS-melting curve analysis in cases scoring negatively, but with strong clinical suspicion [131]. Genome-wide approaches have shown that primary epimutations are focal events involving a 1.6 kb region encompassing the shared *MLH1/EPM2AIP1* promoter [125].

Another LS gene, *MSH2*, is affected by constitutional secondary epimutations. These occur when the 3’ portion of the upstream *EPCAM* gene is deleted, causing transcriptional readthrough within the *MSH2* sequence with production of *EPCAM/MSH2* fusion transcripts and methylation and silencing of the *MSH2* promoter [99]. These 3’ *EPCAM* deletions and associated *MSH2* epimutations account for 1%–3% of the molecular defects identified in LS patients and are conveniently detectable by MLPA in epithelial tissues [132,133]. Patients with tissue-specific mosaic inactivation have a significantly lower incidence of endometrial carcinoma compared to carriers of canonical *MSH2* pathogenic variants [134]. Carriers of *EPCAM* deletions have a 50% chance of passing on the rearrangement to their children, and are managed like typical LS individuals for counseling purposes. 

### 2.4. Neuromuscular Diseases 

#### 2.4.1. Myotonic Dystrophy Type 1

Myotonic dystrophy type 1 (DM1; OMIM#160900) is an autosomal dominant disease characterized by myopathy, progressive muscle weakness, and multisystem complications. The disease results from a CTG repeat expansion (50 to several thousand with tissue-specific differences) in a CpG island of the 3′ untranslated region of the dystrophia myotonica protein kinase (*DMPK*) gene on chromosome 19q13.3 [135]. Several pathogenic mechanisms have been proposed for the repeat expansion including a cis-acting effect reducing *DMPK* gene transcription or translation, an alteration of the chromatin structure at the *DMPK* locus able to repress the expression of neighboring genes, and a toxic gain in the function of the resulting mRNA [136]. Based on the onset of the main symptoms, different clinical subtypes of the disease are recognized, including congenital (CDM1) and childhood onset cases, juvenile and adult onset ones, and late onset DM1 [135]. 

CDM1 is the most severe form and is characterized by large repeat expansions mostly via maternal transmission [137]. Hypermethylation of the region upstream the CTG expansion has been demonstrated in the blood of CDM1 patients [138]. More recent studies revealed that methylation patterns flanking the CTG repeat are stronger indicators of CDM1 than the CTG repeat size, and suggested that *DMPK* methylation may account for the maternal bias of CDM1 transmission, and may serve as a more accurate diagnostic indicator of CDM1 in prenatal screening [137]. Hypermethylation of the upstream region occurs less frequently in DM1, and has been associated with earlier onset of the symptoms, larger CTG expansion and the maternal origin of the expanded allele [139]. In addition, a recent investigation revealed that methylation at the *DMPK* locus in blood DNA contributes significantly and independently of the CTG repeat length to the variability of muscular strength and respiratory profiles in DM1, suggesting that testing for it could improve prognostic accuracy for the patients [140]. 

Several methods can be used to evaluate *DMPK* methylation, including methylation-sensitive high-resolution melting [139,162] or bisulfite sequencing and pyrosequencing approaches [137,140]. The *DMPK* locus contains a binding site for the insulator protein CTCF (CTCF1 site), and the recent models linking DNA methylation to the pathogenesis of DM1 suggest that methylation of this site may inhibit CTCF binding, thus altering the chromatin structure and gene expression of the locus 135,140]. 

#### 2.4.2. Amyotrophic Lateral Sclerosis 

Hexanucleotide GGGGCC repeat expansions in the *C9orf72* gene are the most frequent cause of amyotrophic lateral sclerosis and frontotemporal dementia (OMIM#105550). Expansions of hundreds to thousands of repeats are observed in the patients, but the pathological repeat length threshold is not defined yet [141]. Proposed pathological mechanisms include loss of function of the C9orf72 protein, and the toxic effects of expanded sense and antisense C9orf72 RNA or proteins [141]. Hypermethylation of the *C9orf72* promoter region upstream of the pathogenic repeats was observed in 10–30% of the patients, and linked to increased repeat length and reduced transcription of *C9orf72* [142]. Moreover, the expanded hexanucleotide repeat itself is methylated in almost all cases [143]. However, despite some evidence that intermediate alleles displaying increased *C9orf72* methylation levels are associated with higher frequency of neuropsychiatric symptoms, the clinical utility of *C9orf72* methylation deserves further investigation [144].

#### 2.4.3. Facioscapulohumeral Muscular Dystrophy 

Facioscapulohumeral Muscular Dystrophy (FSHD) is the third most common neuromuscular condition and represents a disease in which genetic defects cause epigenetic changes which result in disease [145]. The pathogenic mechanism is based on overexpression of the retrogene Double Homeobox Protein 4 (*DUX4*) located on the D4Z4 macrosatellite repeat of the chromosome 4q35 subtelomeric region. DUX4 is toxic at high levels, because it activates germline genes, impairs RNA and protein metabolism, and triggers inflammation, oxidative stress and apoptotic events in muscle tissue [146,147]. In most normal adult tissues, D4Z4 is hypermethylated and *DUX4* is turned-off. In FSHD, a disease-specific *DUX4* transcript (*DUX4-fl*) is transcribed from the most distal D4Z4 unit and stabilized by a polyadenylation signal encoded by disease-permissive alleles [148].

Two molecular classes of FSHD associated with similar clinical phenotypes are known. Patients of both classes harbor a *DUX4* allele with a polyadenylation signal. However, 95% of patients (FSHD1) have a deletion of between one and 10 large repeated units of D4Z4 and the rest (FSHD2) inherit a mutation in the *SMCHD1* or *DNMT3B* genes [148,149,150]. *SMCHD1* and *DNMT3B* also act as modifiers of disease severity in FSHD1 subjects [149,151]. Thus, *DUX4* derepression results from D4Z4 chromatin relaxation due to either the contraction of repeats or mutations in chromatin modifiers [148]. Furthermore, 4q35 genes may be epigenetically silenced, because of proximity to the telomere or because of a position-effect of D4Z4, and contraction of D4Z4 repeats may cause their derepression [152]. 

Levels of D4Z4 methylation are negatively correlated with repeat array size, but also with disease severity and disease penetrance. After correction for repeat size, the variability in clinical severity in FSHD1 and FSHD2 individuals depends on individual differences in susceptibility to D4Z4 hypomethylation, with more severe cases showing quantitatively less methylation and less severely affected subjects more methylation [153]. Therefore, the molecular diagnosis of FSHD requires a combination of genetic and epigenetic tests. The evaluation of *D4Z4* methylation levels by bisulfite sequencing in blood or saliva DNA allows discrimination between FSHD1 and FSHD2 [154].

## 3. DNA Methylation Analysis in Prenatal Diagnosis 

So far, the analysis of DNA methylation in prenatal diagnosis pertains only the ImpDis and the triplet expansion disorders FXS and CDM1. Although prenatal diagnosis of FXS and CDM1 is based on the sizing of the triplet repeat, coupling it with DNA methylation analysis may ameliorate diagnostic accuracy. Molecular tests can be offered during the prenatal period, based on the presence of:Abnormal fetal and/or parental karyotypes involving chromosomes harboring imprinted loci;Positive family history;Fetal phenotypes suggesting ImpDis detected by ultrasound;Females carrying PM or FM of FXS and CDM1.

Although not currently recommended, it is expected that requests for prenatal testing of ImpDis in fetuses conceived by assisted reproductive techology (ART) may increase in the future based on the reported association with these procedures [163,164]. An accurate molecular prenatal diagnosis is often challenged by heterogeneity and frequent mosaicism of the genetic/epigenetic alterations [7]. In addition, the offered tests must take into account the source of fetal cells, because methylation patterns of embryonic and extraembryonic tissues may differ from those of adult tissues [165,166]. For these reasons, prior to offering prenatal diagnosis based on methylation testing, a detailed presentation of the technological limitations should be discussed with the parents; in particular, they should be made aware that a normal result does not necessarily exclude the suspected diagnosis. Moreover, for FXS, couples should be informed about the possible identification of a female fetus carrying a FM, who has (approximately) a 50% risk of being clinically affected.

Methylation testing for ImpDis diagnosis can be performed on both chorionic villi samples (CVS) and amniotic fluid (AF). Indeed, germline imprinted DMRs (e.g., *H19/IGF2*:IGDMR and *KCNQ1OT1*:TSSDMR for BWS/SRS and *SNURF*:TSSDMR for PWS/AS) display in the placenta the same methylation levels as in the fetus; in addition, culturing of CVS/AF samples seems not to modify the imprinting pattern of ICs. However, some CpGs of the germline DMRs and the somatic DMRs (e.g. *H19* and *MAGEL2* promoters) could be hypomethylated in CVS and amniocytes, and should not be considered for prenatal diagnosis [165]. For FXS, it is preferable to perform the study in CVS after the twelfth week of gestation, because this locus is not fully methylated at earlier stages. Because of possible epigenetic mosaicism, it is important to verify the sensitivity of the experimental procedures, in order to maximize the diagnostic yield. In addition, false negative results may be related to the mosaic distribution of the epimutation within fetal tissues. At present, prenatal methylation testing is restricted to single disease-specific loci, although it is likely that array- and next generation sequencing-based assays will be implemented in the future. Given the limitations described above, the methods for determining DNA methylation in fetal tissues are the same used for postnatal testing and include MS-MLPA and methylation-specific pyrosequencing combined with molecular karyotyping for determining UPD and CNVs. However, the flowchart for prenatal testing is not necessarily identical to the one indicated for postnatal testing, and may be modified according to known molecular defects and specific clinical features.

## 4. Perspectives and Challenges 

The possibility to diagnose cases with unidentified genetic defects or with variants of uncertain significance and re-classify monogenic disorders on the basis of epigenetic defects of multiple loci is a fascinating opportunity for DNA methylation testing. So far, DNA methylation analysis for diagnostic purposes has been limited to CRDs, some chromosome imbalances, Imprinting Disorders and a small number of other monogenic diseases. However, the number of disorders for which a specific DNA methylation signature can be identified is likely to become larger with the development of more extended arrays and next generation sequencing-based assays, and possibly include other genetic diseases in which DNA binding proteins or their cognate target sites are affected. An important advantage is that disease-specific methylation signatures can be often identified in blood, even if this is not the primary affected tissue, because major methylation reprogramming events occur in early embryogenesis and abnormal patterns may be perpetuated across development and tissues. Another advantage is that DNA methylation is a rather stable modification and is suitable for high throughput testing of multiple loci.

Despite these positive expectations, DNA methylation analysis remains quite challenging. The presence of epigenetic mosaicism and tissue-specific patterns often requires highly sensitive techniques and accurate standardization to perform correct molecular diagnosis. Sophisticated algorithms are sometimes needed to adjust for cell-type composition [157]. It is likely that the emerging single cell methylome analysis will enable us to better dissect cell type-specific patterns and allow identification of even more subtle tissue-specific epimutations [167]. Another common problem is represented by variability due to batch effects in the results of genome-wide analyses, and expert bioinformatics work is needed to obtain reliable results. Finally, methylome analysis will likely lead to the identification of potential new candidates; the challenge will be to identify clinically significant associations. 

## Figures and Tables

**Table 1 genes-11-00355-t001:** Description of DNA methylation abnormalities in mono/oligogenic diseases.

Disease (OMIM)	Chromosome	DNA Methylation Defects	Frequency of Methylation Defects	Associated Genetic Defects	Mosaicism	Recurrence Risk	Methods	Refs.
Fragile X syndrome (300624)	Xq27.3	*FMR1* GOM	100%	Expansion of CGG repeat (>200) in the *FMR1* 5′-UTR	Yes	50% for PM and FM mothers	MS-MLPA	[16,17,18,19,20,21,22,23,24,25]
Claes–Jensen syndrome (300534)	Multiple chromosomes	LOM of 1769 CpGs (9 regions)	100%	*KDM5C* variants	Yes	50% from female carriers to sons	Illumina Infinium BeadChip	[15,26,27,28,29]
Sotos syndrome (117550)	Multiple chromosomes	LOM of >7000 CpGs (1300 regions)	100%	*NSD1* variants	Yes	50%	Illumina Infinium BeadChip	[15,29,30,31]
Kabuki syndrome (147920, 300867)	Multiple chromosomes	LOM of 856 CpGs, GOM of 648 CpGs	100%	*KMT2D* and *KDM6A* variants	Yes	50%	Illumina Infinium BeadChip	[15,29,32,33]
CHARGE syndrome (214800)	Multiple chromosomes	1320 CpGs	100%	*CHD7* variants	Yes	50%	Illumina Infinium BeadChip	[15,29,33]
Alpha thalassemia/mental retardation X-linked syndrome (301040)	Multiple chromosomes	1112 CpGs GOM of 11 regions LOM of 5 regions	100%	*ATRX* variants	Yes	50% from female carriers to sons	Illumina Infinium BeadChip	[15,29,34]
Floating–Harbor syndrome (136140)	Multiple chromosomes	1078 CpGs GOM of 19 regions LOM of 9 regions	100%	*SRCAP* variants	Yes	50% in dominant cases	Illumina Infinium BeadChip	[15,29,35]
BAFopathies (Coffin–Siris (135900, 614608, 614609), Nicolaides–Baraitser (601358) and 6q25 microdeletion (612863) syndromes)	Multiple chromosomes	135–146 CpGs (20–30 regions)	100%	*ARID1B, SMARCB1, SMARCA4, SMARCA2* variants, *ARID1B* deletions	Yes	50%	Illumina Infinium BeadChip	[15,36]
ADNP syndrome (615873)	Multiple chromosomes	LOM of ~6000 CpGs GOM of ~1000 CpGs	100%	*ADNP* variants	Yes	50%	Illumina Infinium BeadChip	[37]
Autosomal dominant cerebellar ataxia with deafness and narcolepsy (604121)	Multiple chromosomes	3562 CpGs (mostly LOM) GOM of 82 regions	100%	*DNMT1* variants	Yes	50%	Illumina Infinium BeadChip	[15,29,38]
Hereditary sensory and autonomic neuropathy type 1 with dementia and hearing loss (614116)	Multiple chromosomes	LOM of 5649 regions GOM of 1872 regions	100%	*DNMT1* variants	Yes	50%	Illumina Infinium BeadChip	[39]
Tatton-Brown–Rahman syndrome (615879)	Multiple chromosomes	LOM of 388 regions GOM of 1 region	100%	*DNMT3A* variants	Yes	50%	Illumina Infinium BeadChip	[40]
Heyn–Sproul–Jackson syndrome (618724)	Multiple chromosomes	GOM of 1140 regions LOM of 738 region	100%	*DNMT3A* variants	Yes	50%	Illumina Infinium BeadChip	[41]
Immunodeficiency-centromeric instability-facial anomalies syndrome 1 (242860)	Multiple chromosomes	LOM of 6942 CpGs GOM of 1921 CpGs	100%	*DNMT3B* variants	Yes	25%	Illumina Infinium BeadChip	[42,43]
Immunodeficiency-centromeric instability-facial anomalies syndrome 2 (614069)	Multiple chromosomes	LOM of 8414 CpGs GOM of 2661 CpGs	100%	*ZBTB24* variants	Yes	25%	Illumina Infinium BeadChip	[42]
Immunodeficiency-centromeric instability-facial anomalies syndrome 3 (616910)	Multiple chromosomes	LOM of 9623 CpGs GOM of 2166 CpGs	100%	*CDCA7* variants	Yes	25%	Illumina Infinium BeadChip	
Immunodeficiency-centromeric instability-facial anomalies syndrome 4 (616911)	Multiple chromosomes	LOM of 8708 CpGs GOM of 4120 CpGs	100%	*HELLS* variants	Yes	25%	Illumina Infinium BeadChip	
Genitopatellar syndrome (606170)	Multiple chromosomes	~700 CpGs	100%	*KAT6B* variants	Yes	50%	Illumina Infinium BeadChip	[15,29]
Say–Barber–Biesecker–Young–Simpson syndrome (603736)	Multiple chromosomes	~800 CpGs	100%	*KAT6B* variants	Yes	50%	Illumina Infinium BeadChip	[15,29]
Werner syndrome (277700)	Multiple chromosomes	LOM of 614 CpGs GOM of 511 CpGs	100%	*WRN* variants	Yes	25%	Illumina Infinium BeadChip	[15]
Williams syndrome (194050)	Multiple chromosomes	1413 CpGs (mostly GOM)	100%	7q11.23 deletions	Yes	50%	Illumina Infinium BeadChip	[15]
7q11.23 duplication syndrome (609757)	Multiple chromosomes	508 CpGs (mostly LOM)	100%	7q11.23 duplications	Yes	50%	Illumina Infinium BeadChip	[15]
Progressive supranuclear palsy (601104)	Multiple chromosomes	GOM of 6110 CpGs LOM of 2818 CpGs	100%	*MAPT* variants	Yes	50%	Illumina Infinium BeadChip	[15]
Frontotemporal dementia (600274)	Multiple chromosomes	LOM of 387 CpGs GOM of 142 CpGs	100%	*MAPT* variants	Yes	50%	Illumina Infinium BeadChip	[15]
Cornelia de Lange syndrome (122470, 300590, 610759)	Multiple chromosomes	GOM of 563 CpGs LOM of 361 CpGs	100%	*NIPBL, SMC1A, SMC3* variants	Yes	50%	Illumina Infinium BeadChip	[44]
SETD1B-related syndrome	Multiple chromosomes	3340 CpGs (mostly GOM)	100%	12q31.24 deletions/*SETD1B* variants	Yes	50%	Illumina Infinium BeadChip	[45]
Prader–Willi syndrome (176270)	15q11.2	*SNURF* GOM	99%	pat deletion of 15q11q13 UPD(15)mat	Yes	<1% for primary epimutations or UPD, 50% for pat deletions	MS-MLPA MS-pyrosequencing	[46,47,48,49,50]
Angelman syndrome (601623)	15q11.2	*SNURF* LOM	80%	mat deletion of 15q11q13 UPD(15)pat	Yes	<1% for primary epimutations or UPD, 50% for mat deletions	MS-MLPA MS-pyrosequencing	[47,49,51,52,53,54,55]
Temple syndrome (616222)	14q32	*MEG3/DLK1* LOM	100%	UPD(14)mat pat deletion of *MEG3/DLK1* IG-DMR chromosomal rearrangements	Yes	<1% for primary epimutations or UPD, 50% for pat deletions	MS-MLPA	[56,57,58,59,60]
Kagami–Ogata Syndrome (608149)	14q32	*MEG3/DLK1* and/or *MEG3* GOM	100%	UPD(14)pat mat deletion of *MEG3/DLK1* IG-DMR chromosomal rearrangements	Yes	<1% for primary epimutations or UPD, 50% for mat deletions	MS-MLPA	[60,61,62,63]
Beckwith–Wiedemann syndrome (130630)	11p15.5–11p15.4	IC2 LOM	80%	UPD(11)pat (up to 10% with whole genome pat UPD) mat IC1 deletion or SNVs mat SCMC SNVs (see MLID) chromosomal rearrangements	Yes	<1% for primary epimutations or UPD, 50% for mat IC1 deletions or SNVs, increased for mat SCMC SNVs	MS-MLPA MS-pyrosequencing	[64,65,66,67,68,69]
		IC2 LOM + IC1 GOM						
		IC1 GOM						
Silver–Russell syndrome (1800860)	11p15.5	IC1 LOM	50%	pat deletion of IC1 UPD(11)mat (rarely whole genome mat UPD) mat SCMC variants (see MLID) chromosomal rearrangements	Yes	<1% for primary epimutations or UPD, 50% for pat IC1 deletions, increased for mat SCMC SNVs	MS-MLPA MS-pyrosequencing	[66,70,71,72,73,74,75,76,77,78,79,80,81,82]
		IC2 GOM + IC1 LOM						
	7	*MEST* GOM + *GRB10* GOM	4–10%	UPD(7)mat (rarely whole genome mat UPD)		<1%		
Pseudohypoparathyroidism 1b (603233)	20q13.32	*GNAS* LOM	100%	mat deletions/duplications of *GNAS* DMRs UPD(20)pat	Not reported	<1% for primary epimutations or UPD, 50% for mat deletions/duplications	MS-MLPA	[83,84,85,86,87,88,89,90,91]
Transient neonatal diabetes mellitus (601410)	6q24	*PLAGL1* LOM	70%	UPD(6)pat chromosomal rearrangements *ZFP57* variants (see MLID)	unknown	<1% for primary epimutations or UPD, 25% with parents carrying *ZFP57* variants	MS-MLPA	[92,93,94]
MLID	Multiple chromosomes	LOM of multiple DMRs	50%–75% of TNDM cases with *PLAGL1* LOM 20%–50% of BWS cases with IC2 LOM 9.5%–30% of SRS cases with IC1 LOM 0%–6.3% of PHP-1b cases with *GNAS* LOM	Maternal-effect SCMC variants Zygotic *ZFP57* variants	Yes	<1% for primary epimutations, Increased in case of maternal-effect SCMC variants or zygotic *ZFP57* variants	MS-MLPA	[95,96,97,98,99,100,101,102,103,104,105,106,107,108,109,110,111,112,113,114,115,116,117,118]
Retinoblastoma (180200)	13q14	*RB1* GOM	13%	Not reported	Yes	<1%	MS-MLPA	[119,120,121,122,123,124]
Lynch syndrome (609310)	3p22.2	*MLH1/EPM2AIP1* GOM	up to 3%	deletions or c.-27C>A and c.85G>T substitutions	Yes	<1% for primary epimutations, 50% in case of genetic alterations	MS-MLPA	[125,126,127,128,129,130,131,132,133,134]
Lynch syndrome (120435)	2p21-p16	*MSH2* GOM	1%–3%	*EPCAM* 3′ deletions	Yes, limited to epithelial tissues	50%	MS-MLPA	
Myotonic dystrophy type 1 (160900)	19q13.3	*DMPK* GOM	100% in the congenital forms 16%–50% in non-congenital forms	Expansion of CTG repeat (>50) in the *DMPK* 3′-UTR	Yes	50% for FM and PM mothers	MS-HRMA bisulphite sequencing	[135,136,137,138,139,140]
Amyotrophic Lateral Sclerosis (105550)	9p21.2	*C9orf72* GOM	10%–30%	Expansion of GGGGCC repeat in the *C9orf72* 5′UTR	Yes	50%	bisulphite sequencing	[141,142,143,144]
Facioscapulohumeral Muscular Dystrophy (158900, 158901)	4q35	D4Z4 LOM	100%	Deletion of D4Z4 repeats (FSHD1) *SMCHD1* and *DNMT3B* variants (FSHD2)	Yes	50% for FSHD1, lower for FSHD2	bisulphite sequencing	[145,146,147,148,149,150,151,152,153,154]

Loss of methylation (LOM); gain of methylation (GOM); uniparental disomy (UPD); full mutation (FM); premutation (PM); methylation-specific (MS); multiplex ligation probe-dependent amplification (MLPA); paternal (Pat); maternal (Mat); Beckwith–Wiedemann syndrome (BWS); Silver–Russell syndrome (SRS); Transient Neonatal Diabetes Mellitus (TNDM); pseudohypoparathyroidism 1b (PHP1b); Multi-Locus Imprinting Disturbances (MLID); Facioscapulohumeral Muscular Dystrophy (FSHD).

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
