# Peer review of "DNA Methylation in the Diagnosis of Monogenic Diseases"

_genes, 2020, doi:10.3390/genes11040355_

Round 1

Reviewer 1 Report

This review is providing informative collections about diagnosis of genetic diseases based on DNA methylation abnormalities. Its systematic and well-organized summaries give a brief introduction for the audience out of epigenetic field. There are few minor concerns for further improvements:

  1. The disturbed methylome of these diseases can be sorted by direct or indirect effects. The mutations/variants of DNMT related diseases result in direct abnormalities, not fully covered here, such as ICF syndrome. Others will be categized as the secondary effects.
  2. Tissue sampling for the diagnosis is missing here. Because the measurement of solid tissues, blood cell, or cell-free DNA are totally different. DNA methylation has its tissues specific pattern. Even the cell type specific pattern makes single cell methylome diagnosis will be the emerging approaches.
  3. Few typos can be found, e.g. Line40, “patters” should be “pattern”.

Reviewer 2 Report

In this review article, the authors summarize the associations between aberrant DNA methylation patterns and various monogenic diseases. This would be helpful for readers to understand the potential mechanisms promoting respective diseases, in addition to pathologic nucleotide changes. Further, the suggested perspectives and challenges define the merits and limitations of DNA methylation analysis for diagnosis. This manuscript would be acceptable for publication. However, there are a couple of concerns that should be addressed before publication, as listed below.

(1)
Too many acronyms/abbreviations are used, and therefore some sentences look like cryptograms. Only important acronyms/abbreviations should be used; for instance, IG for ‘intergenic’ seems to be too much. In addition, I would recommend to add a section of acronyms/abbreviations.

(2)
The section 2.1.2. Chromatin Related Disorders;
=> Somehow, the section 2.1.2 looks low quality writing. It would be better to get some English proofreading.

(3)
Line 138-140, ‘Depending if they introduce…’;
=> The description is not clear.

(4)
Lines 140-142, ‘According to literature, 83 human…71 epigenetic machinery…[26]’;
=> The description looks different from what the reference-26 said. Please check.
